# Colostrum Quality Assessment in Dairy Goats: Use of an On-Farm Optical Refractometer

**DOI:** 10.3390/biology12040626

**Published:** 2023-04-20

**Authors:** Carlos C. Pérez-Marín, David Cano, Francisco A. Arrebola, Valerii H. Petrusha, Pavlo M. Skliarov, José A. Entrenas, Dolores C. Pérez-Marín

**Affiliations:** 1Department of Animal Medicine and Surgery, Campus of Rabanales, University of Cordoba, 14071 Cordoba, Spain; 2Instituto de Investigación y Formación Agraria y Pesquera (IFAPA), Carretera el Viso km 2, Hinojosa del Duque, 14270 Cordoba, Spain; 3Dnipro State Agrarian and Economic University, Serhii Yefremov Str. 25, 49600 Dnipro, Ukraine; 4Department of Animal Production, Campus of Rabanales, University of Cordoba, 14071 Cordoba, Spain

**Keywords:** colostrum, goat, immunoglobulin G, refractometer, Brix

## Abstract

**Simple Summary:**

Newborn goat kids need to consume high-quality colostrum in the first hours after birth to ensure adequate transfer of passive immunity, through the absorption of immunoglobulins. However, some goats fail to produce high-quality colostrum, and tools for on-farm assessment of colostrum quality are needed. This work evaluated the quality of colostrum in Malagueña dairy goats, through measurement of immunoglobulin G (IgG), fat and protein concentrations in colostrum during the first 3 days after parturition. A handheld optical Brix refractometer, an instrument that is easy to use on the farm, was evaluated for accuracy to estimate colostrum quality. The optical Brix refractometer was found to be suitable for on-farm use to estimate the IgG content in the goat colostrum.

**Abstract:**

Failure of passive immunity transfer is one of the main causes of increased susceptibility to infectious agents in newborn kids. To ensure successful transfer of passive immunity, kids need to be fed high-quality colostrum, containing an adequate concentration of IgG. This work evaluated the quality of colostrum obtained in the first 3 days postpartum from Malagueña dairy goats. The IgG concentration in colostrum was measured using an ELISA as a reference method, and it was estimated by optical refractometer. Colostrum composition in terms of fat and protein was also determined. The mean concentration of IgG was 36.6 ± 2.3 mg/mL, 22.4 ± 1.5 mg/mL and 8.4 ± 1.0 mg/mL on days 1, 2 and 3 after parturition, respectively. Brix values obtained using the optical refractometer were 23.2%, 18.6% and 14.1% for days 1, 2 and 3, respectively. In this population, 89% of goats produced high-quality colostrum with IgG concentrations of >20 mg/mL on the day of parturition, but this percentage declined dramatically over the following 2 days. The quality of the fresh colostrum estimated with the optical refractometer was positively correlated with those obtained using ELISA (*r* = 0.607, *p* = 0.001). This study highlights the importance of feeding first-day colostrum to newborn kids and demonstrates that the optical Brix refractometer is suitable for the on-farm estimation of IgG content in colostrum.

## 1. Introduction

In recent years, goat milk production has reached around 10–12% of total milk production in Spain, where it is common to find productive systems based on autochthonous dairy breeds, such as the Murciano–Granadina, Malagueña and Florida breeds. Traditionally, goat kids were kept with their mothers, but numerous farms are nowadays using separation of kids and handfeeding. There are many factors involved in early mortality among newborn kids, such as low birth weight, premature births, multiple births, poor maternal ability and hypothermia. In addition, colostrum intake is the most important factor linked to high survival and growth rates of newborns. The goat placenta prevents the transfer of antibodies from maternal circulation to the fetus, and goat kids are therefore born without any specific immune protection. Absorption of immunoglobulins from colostrum in the first day of life is essential to obtain adequate humoral immunity. In addition, colostrum is a rich source of fat and other nutrients to help ensure survival in the first few days after birth. The composition of colostrum is influenced by factors such as parity, litter size, and breed [1,2,3]. Passive immune transfer depends on the immunoglobulin concentration in colostrum. IgA, IgG and IgM are contained in goat colostrum, but IgG is the most abundant, accounting for approximately about 85% of total immunoglobulins. The ability of the small intestinal enterocytes to absorb IgG from colostrum in the newborn kid is maximal just after birth but declines rapidly and is gone by approximately 24 h of age (gut closure) [4,5]. Whether a kid attains adequate transfer of passive immunity depends on the hours of age at colostrum ingestion, and the mass of IgG ingested, which is a function of both the IgG concentration of the colostrum (quality) and the volume ingested (quantity). Because not all goats produce colostrum with a high concentration of IgG, it is important to be able to measure the quality of colostrum fed to kids or stored in a frozen colostrum bank.

The literature describes different methods for the determination of IgG in goats colostrum such as RIA, electrophoresis, radial immunodiffusion (RID) or enzyme-linked immunosorbent assay (ELISA) [6,7], and on-farm tests to estimate the IgG values, as colostrometers, refractometers or color evaluation [8,9]. Around 24 h after parturition, IgG levels in goat colostrum have been reported to range from 10.6 mg/mL [10] to 73.7 mg/mL [11]. Currently, RID and ELISA tests are the most widely used and accurate methods for measuring the IgG concentration of colostrum [12,13]. Both techniques must be performed in the laboratory and samples are usually frozen prior to analysis. While the aforementioned techniques are precise and sensitive, they are expensive, time-consuming and not feasible for on-farm assessment of colostrum. There are few studies prospecting the use of refractometers to assess colostrum quality in goats, in contrast to the abundant scientific reports carried out in bovines [1,14,15,16,17]. Recently, optical and digital refractometers have been deployed for the estimation of IgG in goat colostrum [10,13,18]; these offer multiple advantages inasmuch as they are low-cost, rapid, easy and practical for on-farm application. Moderate and strong correlation has been reported in goat colostrum for IgG assessment by ELISA and the Brix values (total soluble solids) by refractometry [12,14,17]. However, other studies report no or weak correlation between Brix refractometry values and ELISA when different ELISA assays have been used to measure IgG concentration [13,19]. These differences could be explained by the methodological variations during the IgG measurement or by differences between instruments or ELISA kits used [13,17]. It is reported that colostrum is considered of good quality when it contains equal or higher than 20 mg/mL of IgG [17,18].

The aim of this study was to evaluate the quality of colostrum in Malagueña dairy goats during the first 3 days postpartum, determining the percentage of goats with optimal IgG levels. For this purpose, the ELISA test was used as a reference method to determine IgG content, and the accuracy of a portable optical Brix refractometer was evaluated.

## 2. Materials and Methods

### 2.1. Farm and Management

This study was conducted in Antequera (Málaga, Spain) (36°57′49″ N 4°32′44″ E), in a dairy goat farm with a total of 800 lactating Malagueña breed goats. Reproductive management on the farm is based on natural mating with only one parturition per year, in March. Parturition was monitored daily and the day after birth, the goats were marked and moved to another barn for milking. The kids remained with their mothers for 24 h after birth and they were then separated and relocated to an artificial suckling pen. After separation, the kids were hand fed with colostrum from their mothers for 2 days and then, they were milk fed and housed with older kids.

### 2.2. Experimental Design

A total of 16 goats were involved in this study. Colostrum was collected after parturition (considered as the day 1 sample). In cases where the birth took place at night, colostrum was collected the following morning (10:00–12:00 noon), and in the case of goats kidding during the course of the day, the samples were collected in the late afternoon. Colostrum was also collected from the mothers on the second and third days following parturition (10:00–12:00 noon).

Goats were hand-milked to obtain colostrum. The 2–3 first streams of milk were discarded to avoid the keratin plug and to minimize the risk of bacterial contamination from the teat canal. For collection, 15 mL plastic tubes were inclined 45° under the teat, preventing contact with the udder skin, and around 10 mL of colostrum was obtained from each teat (a total of 20 mL per goat) and mixed. Duplicate samples were obtained, and the tubes were marked for identification purposes.

After collection, fresh colostrum was assessed by an optical refractometer (Zuzi 300, Auxilab, Beriáin, Spain), working with a scale between 0 and 50%, to determine the Brix value. Before the measurement, the refractometer was tared using distilled water. Then, a drop of colostrum was placed on the glass prism and the Brix value was recorded. The analysis was repeated in duplicate. Colostrum was considered to be high in IgG content when the Brix values were greater than or equal to 20 [17]. All the tubes were frozen at −20 °C and stored pending analysis.

Frozen colostrum samples were thawed in a refrigerator (4 °C) for one day prior to assessment. IgG, fat, total protein and Brix were determined in the frozen–thawed samples. IgG content in colostrum was assessed using the goat IgG ELISA kit (Bethyl Laboratories Inc., Montgomery, MD, USA). Colostrum samples were diluted 1:500,000 with dilution buffer using the serial dilution technique, and standard solutions were also prepared. Then, IgG determination was conducted in accordance with the manufacturer’s instructions. Finally, the absorbance was measured at 450 nm using a microplate photometer (Multiskan FC, Thermo Scientific, San Jose, CA, USA). Colostrum was deemed to be of good quality when the IgG level was equal to or higher than 20 mg/mL using the ELISA [17,18]. To determine the Brix value in the frozen–thawed colostrum, samples were mixed and the same procedure as described above for fresh colostrum was applied. Furthermore, to determine the fat and protein contents in frozen–thawed colostrum samples, a milk analyzer (Julie Z9, Scope Electric, Regensburg, Germany) was used.

### 2.3. Statistical Analyses

Data were processed using IBM SPSS Statistics v. 25 software (Chicago, IL, USA). The Shapiro–Wilk test and Levene’s test were carried out to check the normality and homoscedasticity of the data for the studied parameters (Brix, IgG, fat and total protein). Since the data displayed a parametric distribution, ANOVA was carried out to compare differences between day 1, 2 and 3 after parturition. When significant differences were observed (*p* < 0.05), Duncan’s test was applied (a posteriori). The correlations between different parameters were analyzed using Pearson’s test (*p* < 0.001).

A receiver operating characteristics (ROC) curve analysis was applied to estimate the best cut-off point (using the Youden index) [20] for Brix values to differentiate colostrum containing values above or below 20 mg/mL. Fagan’s nomogram was used to estimate the post-test probabilities based on high or low Brix values [21].

Data are expressed as mean ± SEM and differences were considered as significant when *p* < 0.05 for all analyses.

## 3. Results

On day 1 (i.e., shortly after parturition), fresh colostrum exhibited a mean Brix value of 23.2 ± 0.9 [13.0–35.0]. Values decreased significantly (*p* < 0.001) on day 2 (18.6 ± 0.6 (12.0–27.0)) and day 3 (14.1 ± 0.3 (11.0–20.0)). It was observed that 71.8% of the goats had an adequate colostral IgG content based on a cutoff of 20% Brix value. However, this percentage fell to 43.6% on day 2 and only 2.6% of goats showed high IgG values in their colostrum at 3 days postpartum. Brix values obtained for frozen–thawed colostrum are shown in Table 1.

With regard to the IgG concentration of colostrum, the goats presented a mean value of 36.6 ± 2.3 mg/mL (15.7–72.0) after parturition, which decreased significantly (*p* < 0.001) on day 2 (22.4 ± 1.5 mg/mL (9.7–50.0)) and day 3 (8.4 ± 1.0 mg/mL (0–34.8)), as can be seen in Table 1. A total of 88.9% of the kidding goats showed colostral IgG levels above 20 mg/ml at birth, but this percentage was significantly reduced on day 2 (54.3%) and day 3 (5.4%) (Figure 1).

The ability of the refractometer to discriminate optimal or low IgG content in the colostrum samples (using the ELISA test as the reference or gold standard) was evaluated. The area under the curve was 0.806 (0.722–0.891) and the best cut-off point was 18 Brix, showing a sensitivity of 77% and a specificity of 74% (Appendix A).

In this study, the prevalence of low quality colostrum with an IgG concentration of <20 mg/mL by ELISA was 51% over the first 3 days postpartum. When the Brix value was low, the post-test probability result was 77%, as represented by Fagan’s nomogram. However, the post-test probability was 26% when high Brix values were detected (Appendix A).

Fat percentage in colostrum was significantly higher (*p* < 0.001) on day 1 after parturition, showing a mean value of 8% ±0.4 [2.2–13.4], which decreased to 5.5% ±0.4 [1.6–14.9] on day 2 and to 5.1% ±0.4 [1.2–15.5] on day 3 (Table 1). Total protein concentration also showed significant decreases (*p* < 0.001) when comparing day 1 postpartum to day 2 and day 3 (Table 1).

A positive correlation between fresh and frozen–thawed colostrum samples was observed (*r* = 0.971, *p* < 0.001) (Table 2, Figure 2). Fat colostral percentage was also correlated with Brix values both in fresh (*r* = 0.532, *p* < 0.001) and frozen–thawed (*r* = 0.560, *p* < 0.001) samples (Table 2). Moreover, total protein concentration revealed similar results, showing high positive correlation with Brix values both in fresh (*r* = 0.909, *p* < 0.0001) and frozen–thawed (*r* = 0.934, *p* < 0.001) colostrum (Table 2).

A positive correlation between fat and protein content was also observed (*r* = 0.499, *p* < 0.001) (Table 2).

IgG levels, determined by ELISA, showed high correlation with Brix content in fresh (*r* = 0.607, *p* < 0.001) and frozen–thawed (*r* = 0.623, *p* < 0.001) colostrum (Figure 3). IgG colostrum levels were also positively correlated with total protein content (*r* = 0.556, *p* < 0.001), and, to a lesser degree, with fat content (*r* = 0.337, *p* < 0.001) (Table 2).

## 4. Discussion

It is important to ensure the ingestion of high-quality colostrum by newborn goat kids during their first hours of life since it is essential to guarantee immune protection. However, sometimes goat colostrum contains low concentrations of IgG, which is the most important immunoglobulin for goat kids. In this situation, newborns cannot acquire adequate colostral immunity and they exhibit low viability. The problem may be due to the early milking of goats before parturition, insufficient production of immunoglobulins or difficulties in lactation (maternal rejection, weakness of the kids, etc.), among others. Measuring the IgG content in colostrum could therefore help to mitigate the abovementioned problems. Although complex and time-consuming methods, such as ELISA and RID, are available, they are not easy to apply on the farm.

The present study demonstrated that a number of kidding goats have poor quality colostrum based on the low IgG content. This situation is not always related to mortality or weakness of the kids, as newborns can take enough quantity of low-quality colostrum and then absorb sufficient IgGs to provide the required passive immunity [2]. However, the kids that are born from those goats with lower IgG content will be more likely to develop diseases, illness or weakness as a consequence of their incomplete immunity.

The on-farm use of refractometers could partially resolve these drawbacks because they represent a relatively easy, fast and accurate method. This instrument informs about the refraction of the light when it passes through the liquid sample [14]. In the case of colostrum, the high protein content results in increased refraction of light. Recent studies in goats report that optical refractometry is a good option for the assessment of colostrum quality, and no differences are observed compared to digital refractometers [14]. This work enables the establishment of a cut-off threshold for characterizing good-quality colostrum at around 18.5–21.5% [10,13,17]. In other cases, farmers may be interested in determining the colostrum quality of nursing goats—i.e., goats at day 2 or 3 post-partum that are milked to obtain colostrum, which will be frozen to feed orphaned kids. However, only colostrum with a high amount of IgG should be provided to the newborn. It is stated that goat kids need to consume about 10% of their live body weight in colostrum during the first 12–14 h to acquire optimal immunity. It has also been found that goat kids require 4 g/kg of IgG during the first day of life to acquire passive immunization among 80% of animals [18]. The birth weight of Malagueña newborn kids is around 2.46 kg [22], so it is advisable to feed them with about 0.25 L/day of colostrum, divided into four feedings per day. It is thus assumed that a newborn should be fed with at least 12.5 mg of IgG per day in order to acquire an appropriate immunity status, although other components of the colostrum might also be important for the survival of the kids.

In the present trial, the average concentrations of IgG in goat colostrum measured by ELISA on day 1 post-kidding reached 36.6 mg/ml, being lower than those described in some of the previous studies (50.0 mg/mL [18], 54.4 mg/mL [9] or 73.67 mg/mL [11]), but higher than others (10.6 mg/mL [10]). On day 2 and 3 postpartum, IgG values were reduced to 22.4 and 8.4 mg/mL, respectively, in agreement with other published results [10]. A significant reduction in IgG in goat colostrum during postpartum was observed, similar to that previously described using a colostrometer [10]. Our study demonstrates that not all the goats have enough IgG content after parturition, as about 10% of the animals in the present study showed values below the optimal threshold, set over 20 mg/mL according to previous studies [17,18]. Moreover, as more days elapsed, the IgG content was significantly reduced; only 59% of Malagueña goats showed colostrum of adequate quality (based on IgG content) on day 2 after parturition, dropping to 53% on day 3, in consonance with the findings of other authors [11]. The most important period for the transfer of passive immunity from mother goats to newborns is the first 24 h after parturition, the period in which it is imperative for kids to absorb immunoglobulins. A mortality rate in newborn kids has been reported to be around 9.1% [23] and 15% [24] due to low absorption of immunoglobulins. Meanwhile, it was described that colostrum supplementation did not resolve this situation [24,25]; otherwise, the active control of colostrum intake, based on the palpation of the kid’s abdomen and goat’s udder, or on the observation of suckling, could reduce this rate [23].

The use of refractometers to determine colostrum quality is currently highly popular at the farm level. This technique enables the measurement of the total soluble solids and total protein concentration in liquid samples to be ascertained by their refractometry index (Brix index). In the case of colostrum, the Brix value has been associated with IgG content [10,13,18,26]. In the present study, average Brix values were 23.2%, 18.6% and 14.1% on days 1, 2 and 3 postpartum, respectively, broadly in line with other studies [10,13]; other authors have reported a lower Brix value, however [26]. A first area of enquiry focused on ascertaining whether Brix values determined by refractometer in frozen–thawed colostrum showed similar values to fresh colostrum. A high correlation was observed between the Brix degrees recorded in frozen and fresh colostrum (*r* = 0.971, *p* < 0.001), a finding other authors have also reported [14]. In the present study, a careful thawing procedure was conducted, maintaining frozen samples for 24 h in refrigerated conditions. A previous trial conducted in frozen goat colostrum demonstrated that IgG concentration was not affected when samples were thawed at 60 °C in water, under 4 °C in a cold-storage room, at 27 °C (at room temperature) or by microwaves (reaching a final temperature of 55 °C) [7]. A second strand of enquiry focused on whether Brix colostrum values as measured by refractometry could provide accurate information about the IgG content. Good correlation between colostrum IgG content (measured by ELISA) and Brix values (*r* = 0.607 and 0.623 in fresh and frozen colostrum, respectively) was observed. The first study conducted in goats aimed at estimating IgG concentrations (as measured by ELISA) and the Brix values revealed a high correlation (*r* = 0.88) in the Majorera dairy goat breed [18]. Subsequent studies reported a lower correlation in Saanen and Saanen crossbreed goats (*r* = 0.09 and *r* = 0.406, respectively) [10,13]. Colostral IgG evidently varies between breeds [27]. By contrast, IgG values determined by RID and Brix displayed a high correlation (*r* = 0.76) [13]. Although previous studies report inconsistent degrees of correlation between IgG content and Brix values, the present study conducted on the Malagueña breed indicates that Brix refractometers are a viable option for determining the quality of goat colostrum (based on IgG values) on-farm. In addition, previous studies have demonstrated that temperature (5 °C, 20 °C and 38 °C) does not strongly affect the Brix colostrum measurements obtained by a refractometer, which is another advantage for the on-farm colostrum monitoring in comparison with other methods [28].

As described in bovine colostrum, an accurate determination of IgG concentrations could avoid the consumption of low-quality colostrum by newborn kids [29], but for this proposal it is important to establish the appropriate cut-off points for those rapid methods, in correspondence with IgG values higher than 20 mg/mL [17,18]. Previous studies in goat colostrum recommended fixing the cut-off threshold at a Brix value of 19 [13], 20.7 [17] or in the range between 18.5 and 21.5 [10]. However, the present study determined that 18% Brix value is the best cut-off point considering 20 mg/mL of IgG as the reference value, determined by ELISA test. This practice allows the discarding of low quality colostrum on-farm, reducing the incidence of illness, weakness or mortality associated with inadequate colostrum feeding in newborn goats. With the objective to detect colostrum containing low IgG concentrations, it could be interesting to choose a higher Brix cut-off threshold in favor of having higher specificity in the methods. In the present study, to obtain high specificity, around 92.6% and 96.3%, the cut-off point might be fixed around a Brix value of 22–23, which is more in line with others [10,13,17].

This study also evaluated the fat and total protein in colostrum, which were similar to levels previously reported [8,11,30]. Fat levels in colostrum were higher at around 24 h after parturition and, subsequently, declined gradually. Previous studies have reported that colostral fat values are stabilized at day 3 postpartum, exhibiting fat contents comparable to those in normal goat milk [31]. Total protein levels in goat colostrum were also high at parturition and they decreased daily until day 3; at this time, total protein values were kept at values comparable to those of normal goat milk [31,32]. It has been suggested that the high presence of fat and total protein in goat colostrum could be a natural mechanism for ensuring newborns are supplied with sufficient energy to enable them to maintain an adequate corporal temperature. When colostrum is stored to feed orphaned kids, it is important to monitor not only IgG content but also the fat and total protein levels. While refractometers offer accurate information about the total protein in the colostrum samples, as is evidenced in the present study, the same cannot be claimed for fat content. Another study suggested that colostral fat changes are independent of the protein and IgG content variations, but they can affect the Brix value and they can cause the IgG content assessed by a refractometer to be overestimated [29]. Many bioactive proteins such as IgM, IgA, casein, lactoferrin and lactoperoxidase can be found in colostrum, but IgG is the most abundant [33]. Refractometers estimate the total protein in the colostrum and offer a feasible and practical estimation of IgG. In view of the results of the present study, it is suggested that the Brix refractometer represents an appropriate method for estimating the protein content in goat colostrum samples, although other devices, such as a milk analyzer, might be used for selecting good quality colostrum. In broad terms, the colostrum produced on the first day postpartum represents the best option for establishing a colostrum bank due not only to its IgG content, but also to the fat content.

The limitations of this study are related to the small sample size, as it only involved animals from the Malagueña goat breed, which is a specialized dairy breed, and no other breeds were included. However, no similar trials have been previously carried out in this breed, and the findings obtained here could provide new knowledge to dairy goat farmers. The fact that only one commercial farm was involved could also be considered a limitation, as the management practices of this particular farm may have influenced the results. Nevertheless, the data obtained offer an interesting snapshot of this topic, and further studies could be conducted with more animals and farms. Additionally, similar studies could be carried out in meat and/or mixed goat breeds to identify the incidence of low colostrum quality in goats and to improve colostrum feeding management.

## 5. Conclusions

Goat colostrum obtained on the first day after parturition contained the highest concentrations of IgG, with almost 90% of colostrum samples being high quality (IgG concentrations of at least 20 mg/mL). The IgG concentration of colostrum decreased significantly over the first 3 days after parturition, with less than half of colostrum samples being of high quality on day 2, and only 5.4% meeting the 20 mg/mL cutoff on day 3. If the farmer wishes to create a colostrum bank, samples from day 1 after parturition are strongly recommended given that not only IgG but also fat content are present at maximum concentrations. Because not all goats produce adequate quality colostrum to ensure good transfer of passive immunity to kids, it is necessary to check the quality of colostrum samples before freezing in a colostrum bank. This study suggests that the optical Brix refractometer could be considered an appropriate analytical tool to evaluate goat colostrum quality on farms due to the good correlation observed between Brix values and IgG concentration measured by ELISA. A cut-off threshold for the Brix refractometer values of 18% optimized both sensitivity and specificity according to the Youden index. However, to minimize the chance of accidently retaining poor quality colostrum samples for feeding to kids, a higher Brix cut-off of 22–23% is appropriate.

## Figures and Tables

**Figure 1 biology-12-00626-f001:**
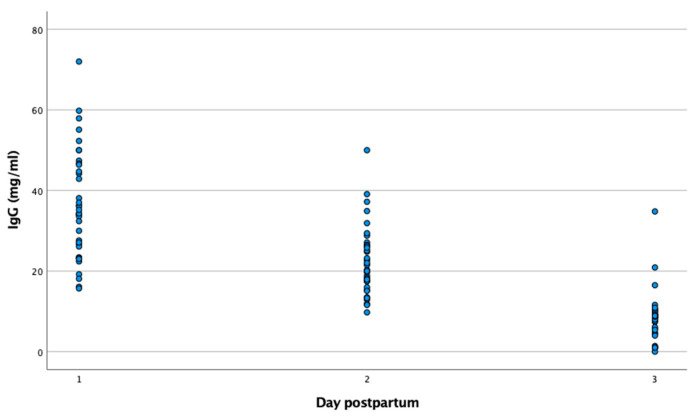
Distribution of IgG concentrations in goat colostrum samples obtained over the first three days after parturition, as measured by ELISA.

**Figure 2 biology-12-00626-f002:**
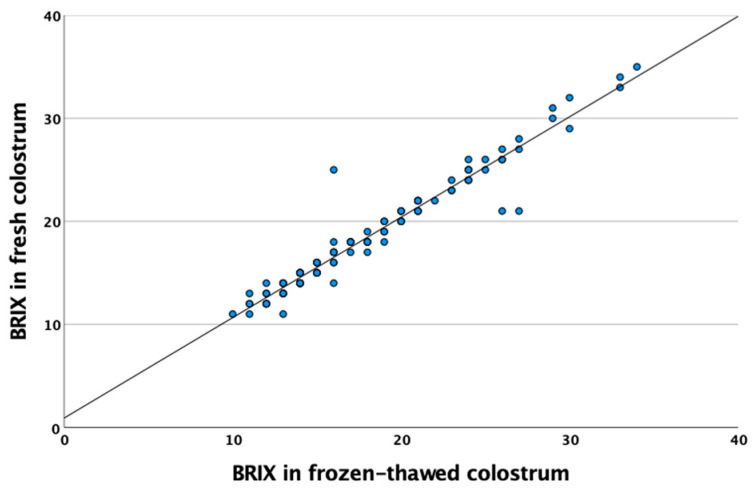
Correlation plots and trend line between fresh and frozen colostrum samples assessed by Brix refractometer.

**Figure 3 biology-12-00626-f003:**
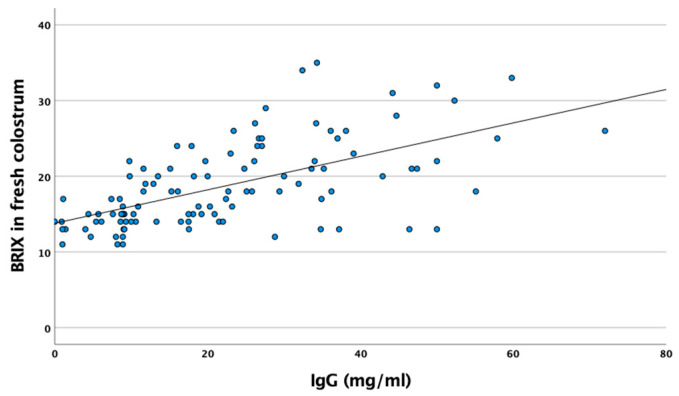
Relation between Brix value measured by optical refractometer and IgG concentration measured by ELISA in fresh colostrum.

**Table 1 biology-12-00626-t001:** Characteristics of goat colostrum collected on days 1, 2 and 3 after parturition.

	Day 1	Day 2	Day 3
**Brix (fresh colostrum) (%)**	23.2 ± 0.9 ^a^	18.6 ± 0.6 ^b^	14.1 ± 0.3 ^c^
**Brix (frozen colostrum) (%)**	22.8 ± 0.9 ^a^	18.2 ± 0.6 ^b^	13.7 ± 0.3 ^c^
**IgG (mg/mL)**	36.6 ± 2.3 ^a^	22.4 ± 1.5 ^b^	8.4 ± 1.0 ^c^
**Fat (%)**	8.0 ± 0.4 ^a^	5.5 ± 0.4 ^b^	5.1 ± 0.4 ^b^
**Total protein (%)**	6.5 ± 0.2 ^a^	5.5 ± 0.2 ^b^	4.3 ± 0.1 ^c^

Different letters (a–c) in the same row indicate significant differences (*p* < 0.001). Data are expressed as mean ± SEM.

**Table 2 biology-12-00626-t002:** Pearson correlation between various colostrum characteristics. Asterisks indicate significant correlation.

	IgG (mg/mL)	Fresh BRIX (%)	Frozen BRIX (%)	Fat (%)	Total Protein (%)
**IgG (mg/mL)**	1	0.607 *	0.623 *	0.420 *	0.572 *
**Fresh BRIX (%)**	0.607 *	1	0.971 *	0.532 *	0.909 *
**Frozen BRIX (%)**	0.623 *	0.971 *	1	0.560 *	0.934 *
**Fat (%)**	0.420 *	0.532 *	0.560 *	1	0.499 *
**Total protein (%)**	0.572 *	0.909 *	0.934 *	0.499 *	1

## Data Availability

The corresponding author will provide the datasets used and/or analyzed during the current work upon reasonable request.

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
