# Peer review of "Colostrum Quality Assessment in Dairy Goats: Use of an On-Farm Optical Refractometer"

_biology, 2023, doi:10.3390/biology12040626_

Round 1
Reviewer 1 Report
I have examined this paper. Couple of issues, the main one relating to novelty. In my view, the current work merely confirms existing knowledge, but I don’t see much / anything that really goes beyond what was already known, or what could have been very logically expected based on what is known. Using Brix for colostrum is known and widely practices, correlation between fresh and frozen is logical (as composition doesn’t change) and the link between IgG and Brix and the changes in composition are also well known. So where is the real novelty? In a way, I think the whole storyline as written in the paper could have been done as well based on existing work and in a sense does not require the data from the study. So, from that perspective, I would really urge authors to find their novelty and make that clearer, rather that for every part having to cite a lot of studies that have reported similar findings already.
Other issue is that introduction and discussion are quite repetitive. They essentially provide the same story, which probably illustrates the previous point on lack of apparent observable novelty at present.
Other points:
-the word ‘quality’ is in many cases mis-used I think. In their work, the authors focus on IgG, which is one aspect of quality. However, quality is multifaceted, so using the term just based on one parameter is not justified. This needs careful rewording throughout the paper
-Line 24: diagnosis strategy for what? Diagnosis is the identification of the cause of an issue or problem. You don’t do that here so the word is not appropriate
-line 27: evaluated, not evaluates
-line 74-75: freezing is not a requirement for ELISA and should not be presented as such. This is just done on (academic) research for sample collection
-Line 101-102: ‘artificially fed’ is not a proper term. Reword (also later instances in the paper)
-Line 115-117: given that milk composition, notably fat content, will change with increasing degree of evacuation of the udder, how representative is a 5 mL sample? And given that 2 samples were taken, how comparable are they?
-Line 129-131: the way this is written suggests that all that is needed is diluting and measuring A450. Clearly not correct
-Line 147 and 149: need references for Youden and Fagan’s
-Line 166: is ‘optimal’ the right word?
-Figure 2, Table 2: really needed? Could be supplementary material I think
-Figure 3: poor quality figure. Looks like a screenshot. Also, could be supplementary
-Line 282: why ‘strikingly’? this would be completely expected I think?
-Line 313-315: do you think you have enough proof to really suggest different recommendations? Sample set is not that extensive and limited to only 1 farm and 1 breed.
Author Response
Reviewer´s comments
(Au) Authors want to expressed our appreciation to the reviewer for its perceptive and constructive criticisms. All their suggestions and recommendations have been considered to improve the manuscript.
Following are listed the answers to the comments and changes are showed in the texts (using changes control).
Reviewer 1
I have examined this paper. Couple of issues, the main one relating to novelty. In my view, the current work merely confirms existing knowledge, but I don’t see much / anything that really goes beyond what was already known, or what could have been very logically expected based on what is known. Using Brix for colostrum is known and widely practices, correlation between fresh and frozen is logical (as composition doesn’t change) and the link between IgG and Brix and the changes in composition are also well known. So where is the real novelty? In a way, I think the whole storyline as written in the paper could have been done as well based on existing work and in a sense does not require the data from the study. So, from that perspective, I would really urge authors to find their novelty and make that clearer, rather that for every part having to cite a lot of studies that have reported similar findings already.
(Au) The present prospective trial is conducted in a commercial farm and implement a simple, useful, rapid tool for determining the colostrum quality in goats. Few studies have been conducted in this specie to compare ELISA test and refractometer, and we want to highlith the importance of determine colostrum quality when a colostrum storage/bank is done. And also, this study offers information about the percentage of goat that show good colostrum quality during the first three days after parturiton, which support the hypothesis that not all the goats that recently give a birth could be used as a colostrum donors. Obviously, this study is based on a practical approach on farm, and this is the added value that we thik that should be consider.
Other issue is that introduction and discussion are quite repetitive. They essentially provide the same story, which probably illustrates the previous point on lack of apparent observable novelty at present.
(Au) Both sections have been reviewed and modified to avoid repetitive sentences.
Other points:
-the word ‘quality’ is in many cases mis-used I think. In their work, the authors focus on IgG, which is one aspect of quality. However, quality is multifaceted, so using the term just based on one parameter is not justified. This needs careful rewording throughout the paper.
(Au) As the referee affirms, "quality" is a wider term. We have modify some parts of the manuscript to avoid misinterpretation. However, in section "2.2. Expermental design", it is clarify that in the present study, colostrum was consider as "good quality" when IgG level was equal or higher than 20 mg/ml using ELISA. Also, it is detailed in "Discussion" section that " poor quality colostrum" is "based on the low IgG content". In any case, all the text has been reviewed and the term "quality" has been removed in some sentences.
-Line 24: diagnosis strategy for what? Diagnosis is the identification of the cause of an issue or problem. You don’t do that here so the word is not appropriate.
(Au) As the reviewer says, the sentence should be corrected.
-line 27: evaluated, not evaluates.
(Au) It has been corrected.
-line 74-75: freezing is not a requirement for ELISA and should not be presented as such. This is just done on (academic) research for sample collection.
(Au) At this point, we try to describe the ussual (not mandatory) way for using ELISA test at farm level. It is not economically profitable to earn money to determine IgG content in 1 or 10 samples of goat colostrum, and then, this determination by ELISA is only (or quite almost) carry out for experimental proposal. In this kind of studies, samples should be frozen because it is not practice to analyse lower than 96 samples, which it is the number of a commercial kit. For this reason, not only in goats, but also in other species, samples are ussualy frozen before the analyses.
-Line 101-102: ‘artificially fed’ is not a proper term. Reword (also later instances in the paper). (Au) The term has been corrected, as suggested by the reviewer.
-Line 115-117: given that milk composition, notably fat content, will change with increasing degree of evacuation of the udder, how representative is a 5 mL sample? And given that 2 samples were taken, how comparable are they?
(Au) Samples were obtained in duplicated. For sampling, a total of 10 ml were obtained from each teat, and they were mixed; then, we got a total of 20 ml of colostrum. Finally, two 5-ml tubes were frozen until analysis. Our decision for colostrum sampling was based in previous studies. In the literature, it is described that samples of 300 ml colostrum was milked (Rudovsky and col, 2007), but others worked with volumes of 50 ml (Zobel and col, 2020), 20 ml (Auad and col, 2016; Buranakarl and col, 2021) or 5 ml (Batmaz and col 2019, Kaçar and col, 2021). The reviewer's observation is really interesting, but we do not know studies analising the IgG concentration in colostrum samples with different volumes.
-Line 129-131: the way this is written suggests that all that is needed is diluting and measuring A450. Clearly not correct.
(Au) This paragraph has been better explained.
-Line 147 and 149: need references for Youden and Fagan’s.
(Au) The following two cites have been added:
Dhamnetiya, D; Jha,R.P.; Shalini, S.; Bhattacharyya, K. How to Analyze the Diagnostic Performance of a New Test? Explained with Illustrations. J. Lab. Physicians 2022, 14, 90–98. 10.1055/s-0041-1734019
Caraguel, C.; Vanderstichel, R. The two-step Fagan’s nomogram: ad hoc interpretation of a diagnostic test result without calculation. Evid. Based Med. 2013, 18, 125–128. 10.1136/eb-2013-101243
-Line 166: is ‘optimal’ the right word?
(Au) These words has been deleted.
-Figure 2, Table 2: really needed? Could be supplementary material I think.
(Au) Both have been moved to Supplementary files.
-Figure 3: poor quality figure. Looks like a screenshot. Also, could be supplementary.
(Au) The quality of the figure 3 has been improved and it will be moved to Supplementary file.
-Line 282: why ‘strikingly’? this would be completely expected I think?
(Au) In agreement with the reviewer, this adjective is not apropriated. It has been removed.
-Line 313-315: do you think you have enough proof to really suggest different recommendations? Sample set is not that extensive and limited to only 1 farm and 1 breed.
(Au) Limitations of this study have been included in Discussion section.
Reviewer 2 Report
Please see attached file

Author Response
Reviewer´s comments
(Au) Authors want to expressed our appreciation to the reviewer for its perceptive and constructive criticisms. All their suggestions and recommendations have been considered to improve the manuscript.
Following are listed the answers to the comments and changes are showed in the texts (using changes control).
Review 2 of Biology manuscript biology-2261082
Colostrum quality assessment in dairy goats: use of optical refractometer on farm
Carlos C. Pérez-Marín, David Cano, Francisco Antonio Arrebola, Valerii H. Petrusha, Pavlo M. Skliarov, Jose Antonio Entrenas, Dolores Pérez-Marín
Comments for Authors:
Simple Summary
Lines 16-24: The whole simple summary needs rewording slightly for English language improvements. My suggestion is as follows, or something similar:
“Newborn goat kids need to consume high-quality colostrum in the first few hours after birth to ensure adequate transfer of passive immunity, through the absorption of immunoglobulins. However, some goats fail to produce high-quality colostrum, and tools for on-farm assessment of colostrum quality are needed. This work evaluated the quality of colostrum in Malaguena dairy goats, though measurement of immunoglobulin G (IgG), fat, and protein concentrations of colostrum during the first 3 days after parturition. A handheld optical Brix refractometer, an instrument that is easy to use on farm, was evaluated for accuracy to estimate colostrum quality. The optical Brix refractometer was found to be suitable for on farm use to estimate the quality of goat colostrum.”
(Au) This paragraph has been modified in the light of the reviewer's comment.
Abstract:
Line 25: Change to “Failure of transfer of passive immunity…..”
(Au) It was changed.
Lines 26-27: Change to: “To ensure successful transfer of passive immunity, kids need to be fed high-quality colostrum, containing an adequate concentration of IgG.”
(Au) It was changed.
Lines 27:28: Change to: “This work evaluates the quality of colostrum obtained in the first 3 days postpartum from Malaguena dairy goats. IgG concentration in colostrum was measured using an ELISA as a reference method……”
(Au) It has been modified.
Line 30: Please add SEM to means .
(Au) These values has been added.
Line 31: Change the sentence that starts with “The colostrum quality….” to “Brix values obtained using the optical refractometer were 23.2%, ……..”, and also add SEM values here. (Au) This sentence was corrected.
General comment: Given that one of your major aims of the study, and the main item you focus on in the conclusions is the proportion of goats producing adequate quality colostrum on each of the days postpartum, I think you should include at least the day 1 data in the abstract. Maybe “In this population, 89% of goats produced high-quality colostrum with IgG concentrations of >20mg/ml on the day of parturition, but this percentage declined dramatically over the following 2 days.” or something similar. If you do that you could take out the first part of the sentence that starts on line 34 as follows:
Line 34: Change to: “This study highlights the importance of feeding first-day colostrum to
newborn kids, and demonstrates that the optical Brix refractometer is suitable for the on farm
estimation of colostrum quality.
(Au) This suggestion was added to the text.
Introduction:
General comment: I feel like your introduction needs quite a bit of work; a lot of this seems to stem from issues with translation to the English language. I’m listing my suggestions below.
Lines 45-46: artificial lactation is not the correct term. You could say “artificial feeding” or “separation of kids and hand-feeding”.
(Au) It has been corrected.
Lines 49-52: Combine these sentences to read: “The goat placenta prevents the transfer of antibodies from maternal circulation to the fetus, and goat kids are therefore born without any specific immune protection”. Remove the last part of the sentence on line 51-52.
(Au) It was modified.
Lines 52-56: Change these sentences to read as follows, or something similar: “Absorption of immunoglobulins from colostrum in the first day of life is essential to obtaining adequate humoral immunity. In addition, colostrum is a rich source of fat and other nutrients to help ensure survival in the first few days after birth. The composition of colostrum is influenced by factors such as parity, litter size, and breed.”
(Au) Suggestions were added.
Line 56: It seems that here you should add information here about the amount of variation reported for colostrum IgG concentrations in goats, as that is part of the justification for undertaking this study (i.e. how do you know that not all goats have excellent quality colostrum). (Au) The suggested explanation was added at the end of that paragraph.
Lines 59-66: It seems unnecessary to go into the mechanisms of IgG absorption in the small intestine in this introduction; your study doesn’t look at kids at all. I think you could take out these sentences. If you feel strongly that you need to cover hours of age at first feeding and its relationship to FTPI maybe you could just say: “The ability of the small intestinal enterocytes to absorb IgG from colostrum in the newborn kid is maximal just after birth, but declines rapidly and is gone by approximately 24 hours of age (gut closure). Whether a kid attains adequate transfer of passive immunity depends on the hours of age at colostrum ingestion, and the mass of IgG ingested, which is a function of both the IgG concentration of the colostrum (quality) and the volume ingested (quantity). Because not all goats produce colostrum with a high concentration of IgG, it is important to be able to measure the quality of colostrum fed to kids or stored in a frozen colostrum bank.”
(Au) Suggestions were added.
Line 67: Remove the sentence that starts with “IgG content is used….”
(Au) It was modified.
Line 67: Add “The” in front of “literature”
(Au) It was modified.
Line 70: Use “on farm” instead of “other faster”
(Au) It was modified.
Line 73: Change sentence to read: “…….most widely used and accurate methods for measuring the IgG concentration of colostrum.”
(Au) It was modified.
Line 76: Change sentence to read: “….not feasible for on farm assessment of colostrum quality.”
(Au) It was modified.
Line 81-82: Remove the sentence starting “But these methods are required….”
(Au) It was removed.
Line 84: Use “Brix values” not “Brix content”, and remove the word content after total soluble solids.
(Au) It was corrected
Line 85: I think this sentence is somewhat misleading, at least regarding reference 13. In that study they had good correlation between the Brix refractometer values and IgG concentration by RID assay (their gold standard), but the ELISA they used did not seem to be suitable for colostral IgG measurement as it did not correlate with either the RID or the Brix. I think there are an adequate number of studies out there now (which you have included in your references) that we can be confident that optical refractometry correlates with colostral IgG in goats. Maybe you could say: “However, other studies report no or weak correlation between Brix refractometry values and ELISA when different ELISA assays have been used to measure IgG concentration.” Or just take this sentence out.
(Au) The suggested sentence has been included.
Line 88: It would be helpful to introduce the cutoffs for IgG concentration that have been recommended to represent good quality colostrum (one or two sentences).
(Au) A sentence has been included.
Line 92: Instead of “feasibility” use “accuracy of a portable optical Brix refractometer…”
(Au) It was modified.
Materials and Methods:
The methods section is reasonably well written and comprehensive. My biggest issue with the study design is that day 1 colostrum was not always collected immediately after birth, and you did not report what proportion of does had been suckled by kids prior to day 1 colostrum samples being collected. I understand the practical reasons for sampling the way you did (it simulates what is likely to be done on farm) but I think it would have been important to report the % of does that had been suckled prior to colostrum collection and to examine whether that was associated with colostrum IgG concentration on day 1. If you have that data I think it would improve the paper to add it. Because the study was only using one breed of goat on one farm, it is important to discuss this limitation in the discussion section.
(Au) Data required by the reviewer were not obtained and then, they can not be supplied. It will be consider for further studies, but it is not possible now. Limitations were included at the end of "Discussion" section.
Lines 101-102: Change to “….artificial suckling pen. After separation, the kids were hand fed colostrum from their mothers…..”
(Au) It was corrected.
Line 107: Change “In case the birth….” to “In cases where the birth…”
(Au) It was modified.
Lines 109-110: Remove the sentence starting “While some goats…..”, it does not add anything to the paper.
(Au) The sentence was removed.
Lines 120-121: Remove “(total soluble solids content)”, it is not necessary.
(Au) It was removed.
Line 124: You use reference #10 (Buranakarl et al) for the Brix cutoff of 20, but when I looked at this paper it seems like they used cutoffs of 18.5% or 21.5%, depending on whether they wanted to optimize specificity or sensitivity. They reference the cutoff of 20.7% used by Kessler (reference #17); please change to reference 17 or add it here, unless I’m missing something!
(Au) Reference Brix cut-off thresholds were corrected.
Line 133: Add “using the ELISA” after 20 mg/ml.
(Au) It was added.
Line 133: Add “value” after Brix
(Au) It was added
Results:
Throughout: Please add % after Brix values (e.g. Line 154 should read 23.2 +/- 0.9%.....)
(Au) It was added.
Line 156: Please add “based on a cutoff of 20% Brix value.” after “….colostral IgG content”
(Au) It was added.
Tables: Please add description of data type to footnotes or description of tables where relevant. For example, Table one needs a footnote to say that data are expressed as mean +/- SEM (the reader shouldn’t have to go back to the methods to find how data are expressed in the tables). (Au) This sentences was added as a footnote.
Line 162: Change “…IgG levels….” to “IgG concentrations” or “the IgG concentration of…”
(Au) It was corrected.
Line 164: Remove “the” before “Table 1.”
(Au) It was corrected.
Line 169: Figure 1 Legend Change to: “Distribution of IgG concentrations in goat colostrum samples obtained over the first three days after parturition, as measured by ELISA.” .”
(Au) It was modified.
Line 178: Figure 2 Legend, change to read “…..using an optical Brix refractometer to estimate the quality of fresh goat colostrum samples” .”
(Au) It was corrected.
Line 181: Table 2 legend: Add “measured by ELISA” to the end of the Table description.
(Au) It was added.
Table 2: Add “% Brix” to the Cut-off value column or list as a footnote.
(Au) It was modified.
Line 184: Change to read “ In this study, the prevalence of low quality colostrum with an IgG concentration of <20 mg/ml by ELISA was 51% over the first 3 days postpartum.”
(Au) It was modified.
Line 187: I would take Figure 3 out – it doesn’t add much to the paper and is described adequately in the text.
(Au) Figure 3 has been moved to Supplementary files.
Lines 191-200: Please use concentration or % instead of content when referring to fat and protein
(Au) It was corrected.
Line 203: Use “various” instead of “different”
(Au) It was corrected.
Discussion:
In general, the content of the discussion section is appropriate and fairly well done. I have not edited the discussion section, but it would be helpful if the authors had someone with English as their first language provide edits in this section. Please use “concentrations” rather than “levels” when referring to IgG.
(Au) It was modified.
It would be helpful to add some discussion on the limitations of the present study regarding external validity, for example, discuss the fact that this study was only conducted in one herd of Malaguena goats and why you think that it is still appropriate to extrapolate to goats in general (I’m not saying it’s not appropriate, just that I think you should justify this in the discussion section as it is an important limitation of your study).
(Au) Limitations were described in the manuscript, at the end of the "Discussion" section.
Lines 308-312: Repeated information; you have already mentioned several times that refractometers are potentially a fast and accurate method for on-farm estimation of colostrum quality – you could probably significantly shorten or remove the sentence starting with “Brix method is attracting…..” .
(Au) The sentence was removed.
Conclusions:
There was little focus on the Brix data in the conclusions, and this section was difficult to read. Suggestions are:
Lines 349-351: “Goat colostrum obtained on the first day after parturition contained the highest concentrations of IgG, with almost 90% of colostrum samples being high quality (IgG concentrations of at least 20 mg/ml). The IgG concentration of colostrum decreased significantly over the first 3 days after parturition, with less than half of colostrum samples being of high quality on day 2, and only 5.4% meeting the 20 mg/ml cutoff on day 3.”
(Au) This sentence was added.
Line 354: If you make the change suggested above, remove the sentence starting “However, less than half….”
(Au) It was removed.
Line 355: Instead of “For the reasons mentioned…..” start this sentence with “Because not all goats produce adequate quality colostrum to ensure good transfer of passive immunity to kids, it is necessary….”
(Au) It was corrected.
Line 357: Use “optical Brix refractometer” instead of just “refractometer”
(Au) It was corrected
Line 359: Use “concentration measured by ELISA” instead of “assessment”
(Au) It was corrected
Lines 359-362: This was an awkward finish to your paper. You should reword these last 2 sentences. A suggestion is “A cut-off threshold for the Brix refractometer values of 18% optimized both sensitivity and specificity according to the Youden index. However, to minimize the chance of accidently retaining poor quality colostrum samples for feeding to kids, a higher Brix cut-off of 22-23% is appropriate.”
(Au) This sentences were changed.
Reviewer 3 Report
Congratulations to the authors for this very interesting study on the use of colostrum in goats. They have done very practical recommandations, very usefull for the farmers and practitionners.
Some specific remarks :
Line 45 : could you give a figure do fix the proportion of farms using artificial milk ?
Line 101 The and not the
Line 102 what kind of colostrum is given at the second day if the kids are separated from their mothers after the first day ?
Line 251 : mg or g ?
Author Response
Reviewer´s comments
(Au) Authors want to expressed our appreciation to the reviewer for its constructive criticisms. All their suggestions and recommendations have been considered to improve the manuscript.
Following are listed the answers to the comments and changes are showed in the texts (using changes control).
Reviewer 3
Congratulations to the authors for this very interesting study on the use of colostrum in goats. They have done very practical recommandations, very usefull for the farmers and practitionners.
Some specific remarks :
Line 45 : could you give a figure do fix the proportion of farms using artificial milk ?
(Au) No information was obtained in reference to the percentage of dairy goat farms using artificial feeding for kids.
In: "HM Vickery, RA Neal, RK Meagher. Rearing goat kids away from their dams 1. A survey to understand rearing methods. Animal 2022, 16:100547". Information on artificial milk feeding systems is limited and currently focuses on large-scale commercial farms within the dairy industry, however, kids are reared artificially in non-dairy systems for a multitude of reasons including those related to management, rejection by the dam, multiple births, and mastitis. Despite the prevalence of goat kids being reared artificially, little research exists surrounding optimal milk intakes and methods of feeding milk after the colostrum feeding stage (last reviewed by Lu and Potchoiba, 1988).
Line 101 The and not the.
(Au) It has been corrected.
Line 102 what kind of colostrum is given at the second day if the kids are separated from their mothers after the first day ?
(Au) It was supplied around 1 liter per day split in four times, that it is considered as ad libitum colostrum suplementation. It is included into the text.
Line 251 : mg or g ?
(Au) The correct unit is gram; It has been corrected.
Round 2
Reviewer 1 Report
Comments have been addressed OK